# One year prevalence of psychotic disorders among first treatment contact patients at the National Psychiatric Referral and Teaching Hospital in Uganda

**Emmanuel Kiiza Mwesiga**[1,2,3]*, **Noeline Nakasujja**[1,2], **Juliet Nakku**[4], **Annet Nanyonga**[4], **Joy Louise Gumikiriza**[1,2], **Paul Bangirana**[1,2], **Dickens Akena**[1,2], **Seggane Musisi**[1,2]

1 Department of Psychiatry, College of Health Sciences, Makerere University, Mulago, Uganda,
2 NURTURE Mental Health subgroup, College of Health Sciences, Makerere University, Mulago, Uganda,
3 Department of Psychiatry and Mental Health, University of Cape Town, Cape Town, South Africa,
4 National Psychiatric Teaching and Referral Hospital of Uganda, Kampala, Uganda

* emwesiga@chs.mak.ac.ug

**Data Availability Statement:** The minimal anonymized data set necessary to replicate our

## Abstract

### Introduction

Hospital based studies for psychotic disorders are scarce in low and middle income countries. This may impact on development of intervention programs.

### Objective

We aimed to determine the burden of psychotic disorders among first treatment contact patients at the national psychiatric referral hospital in Uganda.

### Methods

A retrospective patient chart-file review was carried out in March 2019 for all patients presenting to the hospital for the first time in the previous year. Patients were categorised into those with and without psychotic disorders. We collected sociodemographic data on age, gender, occupation, level of education, ethnicity, religion and home district. We determined the one year prevalence of psychotic disorders among first treatment contact patients. Using logistic regression models, we also determined the association between psychotic disorders and various exposure variables among first treatment contact patients.

### Results

In 2018, 63% (95% CI: 60.2–65.1) of all first time contact patients had a psychosis related diagnosis. Among the patients with psychotic disorders, the median age was 29 years (IQR 24–36). Most of the patients were male (62.8%) and unemployed (63.1%). After adjusting for patients' residence, psychotic disorders were found to be more prevalent among the female gender [OR 1.58 (CI1.46–1.72)] and those of Pentecostal faith [OR 1.25 (CI 1.10–1.42)].

study findings has been uploaded as Supporting Information files.

**Funding:** The work was supported by Grant Number D43TW010132 supported by Office Of The Director, National Institutes Of Health (OD), National Institute Of Dental & Craniofacial Research (NIDCR), National Institute Of Neurological Disorders And Stroke (NINDS), National Heart, Lung, And Blood Institute (NHLBI), Fogarty International Center (FIC), National Institute On Minority Health And Health Disparities (NIMHD). Its contents are solely the responsibility of the authors and do not necessarily represent the official views of the supporting offices.

**Competing interests:** The authors declare no competing interests.

## Conclusion

Among first treatment contact patients in Uganda, there is a large burden of psychotic disorders. The burden was more prevalent among females as well as people of Pentecostal faith who seemed to use their church for faith-based healing. Incidence studies are warranted to determine if this phenomenon is replicated at illness onset.

## Introduction

Psychotic disorders that include schizophrenia spectrum disorders as well as bipolar affective disorders are the leading contributors to disease burden globally [1–3]. Schizophrenia was assigned the highest disability coefficient in global burden of disease (GBD) study [4, 5]. Psychotic disorders run a chronic course in the life of an individual. They usually present in early adolescence with a first episode of psychosis; and then continue with some form of disability thorough out the life of the individual [6]. Patients with psychotic disorders are more likely to have worse social functioning, poor quality of life and die earlier than their peers [7–12]. Correct management at initial presentation of psychotic disorders -operationally defined as the first episode of psychosis (FEP); has been associated with lower relapse rates, greater functional recovery and improved quality of life [13, 14]. Worldwide the prevalence for psychotic disorders has remained relatively stable between 1–3% even in low and middle income countries (LMIC) like Uganda [3]. Hospital based prevalence rates for psychotic disorders especially among first time attended in LMIC are however scarce. The current literature in the Ugandan setting has mainly dwelt on people with HIV/AIDS among first time mental treatment contacts [15].

There is limited literature on the burden of psychotic disorders at initial mental treatment contact in LMICs [16]. It is unclear if the burden of psychotic disorders is greater than that for other disorders like anxiety, mood or substance use disorders. In a previous review by Steel et al 2014, the period prevalence of common mental disorders like anxiety, mood and substance use disorders was found to be lower in low resource countries than high income countries [17]. Such information is crucial in human resource allocation and the development of specialised services in tertiary care. In The sociodemographic profile of patients presenting to tertiary care in the Ugandan setting is not well described. For example, literature has shown higher incident rates for psychotic disorders among males than females [18–22]. Whether this is replicated at presentation for care in our setting is unknown. Also, the clinical profiles of the various psychotic disorders are unknown. This is especially important as management differs between the different psychosis spectrum disorders [23]. The majority of patients with psychotic disorders prefer alternative and complimentary therapies over western medicine [24–30]. It is unclear if this preference translates to lower rates and/or different clinical profiles for psychotic disorders among patients presenting to mental health services for the first time. Such differences are important in directing policy and developing interventions to improve care for patients with psychotic disorders.

Describing the burden and risk factors for psychotic disorders at initial treatment contact is a crucial step in developing interventions to improve the outcomes for patients with psychotic disorders. In Uganda there is a precedent for this approach where extensive literature on the burden of HIV/AIDS in the psychiatric setting was instrumental in development of interventions for patients with severe mental illness suffering with AIDS [31–35]. The current study therefore aims to determine the burden of psychotic disorders among initial treatment contact patients at the national psychiatric hospital in Uganda.

## Methods

The study took place at Butabika National Psychiatric Referral and Teaching Hospital, a 600 bed capacity mental hospital located approximately twelve kilometres from Kampala [36]. The hospital is located in the heart of the Greater Kampala Metropolitan Area (GKMA) where 10% of Uganda's population reside and responsible for a third of the country's gross domestic product (GDP) [37]. Butabika National Psychiatric Referral and Teaching Hospital determines the policy agenda for mental health in the country together with the Ministry Of Health and is responsible for various levels of mental health training [38]. It also plays a supervisory role over all mental health provision services in the country that include 12 regional referral hospitals and 96 district hospitals. Functioning below the district hospitals are three different levels of health centres (HC) namely HC4, HC3 and HC2. Mental health provision starts at HC3 level with subsequent referrals to higher centres. Currently, the hospital has specialised services for substance use disorders at the Alcohol and drug unit, a forensic ward, a specialised child and adolescent mental health unit as well as specialised occupational therapy and psychotherapy units. In terms of human resource allocation, the national psychiatric and teaching hospital is run by 72 clinicians (psychiatrists' clinical psychologists and psychiatric clinical officers); 157 nurses, 4 social workers and 59 mental attendants. Given that it is a national referral hospital it also provides non psychiatric care like HIV/AIDS care, minor surgeries and dental services. Like in many similar facilities in LMICs there are a number of challenges in provision of services primarily due to limited budgetary allocation [38, 39].

We used a retrospective case analysis of chart records to determine the burden, profile and associated factors for psychotic disorders among first treatment contact patients. Approval for the study was obtained from the Uganda National Council for Science and Technology (UNCST) and the School of Medicine Research and Ethics Committee (SOMREC) of Makerere University. We also received institutional approval from the hospital to carry out the study. As this was a retrospective chart review of file records, we did not receive patient consent. All patients presenting to the hospital for the first time who had a psychiatric diagnosis on file between January 1st and December 31st, 2018 made our study population. We excluded patients presenting for the first time for non-psychiatric services like dental services, routine HIV care or minor surgeries like circumcision.

On a routine clinic day, the hospital records team opens a file for all patients presenting to the hospital for the first time. The patient sociodemographic variables including age, gender, ethnicity, religion, occupation and home district are recorded in the file before the patient is sent to see a clinician. These were the sociodemographic characteristics abstracted for in the chart review. The clinician then makes a diagnosis, and a decision of whether to treat the patient as an out-patient or send them to admission in one of the units described above. Diagnoses as well as criteria for severity are made based on the Diagnostic and Statistical Manual of Mental disorders 5th edition (DSM-5) [3]. Once the patient has received care, the health care workers return the patient file to the records office for safe storage. Some patients receive care as in-patients, and others are treated as out-patients and return to their homes the same day.

We used standardized questionnaires to extract sociodemographic and diagnosis data from the chart files of all patients presenting to the hospital for the first time from January to December 2018. The files reviewed were for participants who presented for the first time at the hospital irrespective of whether they were outpatients and returned home or admitted as in-patients for further management. Diagnoses of schizophrenia spectrum and related psychoses, bipolar affective disorder and mood disorders with psychotic disorders were classified as psychotic disorders. As these participants with psychotic disorders were presenting to the clinic for the first time they were classified as having a first episode of psychosis (FEP). All other

diagnoses among patients presenting for the first time including but not limited to temporal lobe epilepsy, anxiety disorders, substance use disorders and depressive disorders were classified as non-psychotic disorders. We considered sociodemographic characteristics as the exposure variables and the diagnostic categories as the outcome variables. Abstracted data from the files was entered into Epidata 3.1 by a database manager and exported to Stata version 13 for analysis. Data analysis was conducted in March 2019.

Proportions of patients by different diagnostic categories were calculated to determine the one year prevalence of psychotic disorders. Using bivariate analysis, we compared the proportions of participants with psychotic disorders to non- psychotic disorders along various exposures. As the age of the participants was skewed, this was recoded to those with ages less than or equal to the median and those greater than the median. No variables exhibited any collinearity and the dataset had no outliers. We used a modified Poisson regression model to establish factors associated with psychotic disorders given that it has robust standard errors and therefore gives more accurate confidence intervals. Variables with a level of significance less than 0.2 were included in the bivariate analysis. However, region of origin was assessed for any possible confounding effects as ethnicity has been shown to have a genetic biological risk factor for psychotic disorders. At multi-variate analysis a level of significance of less than 0.05 was used to test for significance between different exposures and FEP.

## Results

Between January 1st, 2018 and December 31st, 2018; 43,870 patients were seen in the outpatient mental health clinic and 10,578 patients were admitted with a bed occupancy rate of 149%. 1685 patients accessed services from Butabika for the first time and made up the study sample. A total of 201 (11.93%) patients lacked a diagnosis in their records and were excluded from the final analysis. The total number of records reviewed for this study was 1484. On average there were 5 new patients each day accessing the hospital for the first time during the year 2018. Fig 1 shows the proportions of patients seen by month and gender. There was wide representation of different ethnicities in the sample with over 25 different tribes presenting to the hospital. Other baseline characteristics of all new participants are highlighted in Table 1. Among all new patients, the commonest diagnosis was a non-affective psychosis accounting for 32.01% of the total sample closely followed by substance use disorder at 30.39%. Anxiety disorders were the least common final diagnosis at 0.47%. The frequencies of different diagnoses among the total sample are highlighted in Fig 2.

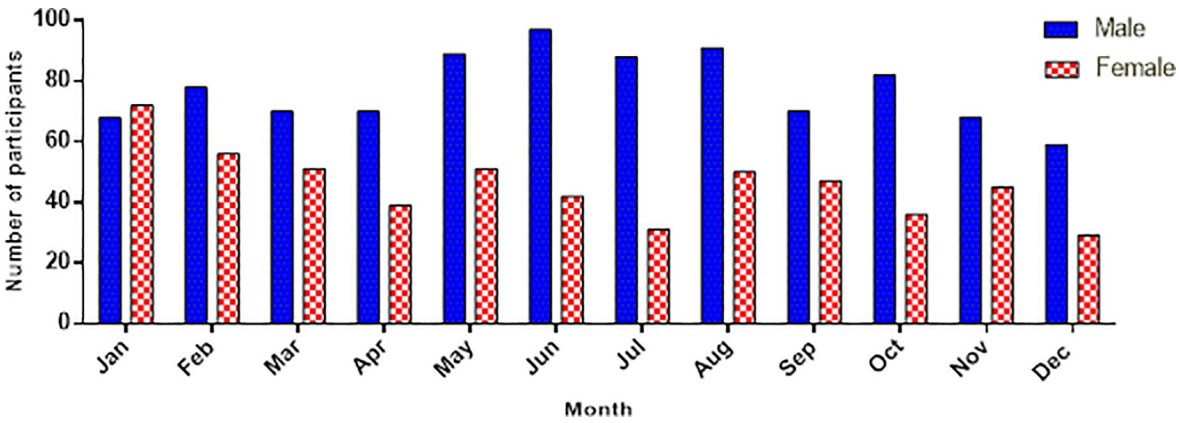

**Fig 1. Bar graph of number of participants by month of the year and gender.**

**Table 1. Background characteristics of all patients who reported for the first time in 2018.**

| Variable | Number (N) | Percentage (%) |
|---|---|---|
| **Age** | | |
| 18–24 | 459 | 28.8 |
| 25–29 | 377 | 23.6 |
| 30–36 | 371 | 23.3 |
| 37–47 | 231 | 14.5 |
| 48–53 | 81 | 5.1 |
| 54–86 | 77 | 4.8 |
| **Gender** | | |
| Male | 930 | 62.8 |
| Female | 549 | 37.1 |
| **Religion** | | |
| Protestant | 404 | 32.3 |
| Catholic | 407 | 32.5 |
| Moslem | 242 | 19.3 |
| Seventh day Adventist | 29 | 2.3 |
| Pentecostal/Born again | 123 | 9.8 |
| Other religions | 46 | 3.7 |
| **Occupation** | | |
| Student | 89 | 6.4 |
| Formal | 108 | 7.8 |
| Non-formal | 270 | 19.4 |
| Unemployed | 922 | 66.4 |
| **Region** | | |
| Central | 1,093 | 79.9 |
| Eastern | 102 | 7.5 |
| Northern | 30 | 2.2 |
| Western | 143 | 10.5 |

## Burden of psychotic disorders

Approximately two-thirds [62.7% (95% CI: 60.2–65.1)] of all patients had a psychotic disorder. Among the patients classified as having psychotic disorders, 51.08% were classified as having schizophrenia spectrum disorders, 30.75% as bipolar affective disorders and 18.17% as an organic psychosis. The median age for patients with psychotic disorders was 29 years (IQR 24–36) with almost twice as many males as females. Most participants (76.03%) were between the 30 to 39 age range with only 4.54% of patients below the age of 18 years. Other baseline characteristics of the patients with psychotic disorders are shown in Table 2.

At bi-variate analysis, psychotic disorders were found to be more prevalent among the female gender [Prevalence ratio (PR) 1.54 (confidence interval 1.43–1.66)] as well as patients who reported to subscribe to the Catholic [PR 1.13 (CI 1.01–1.26)] or Pentecostal faiths [PR 1.36 (CI 1.19–1.54)]. Psychotic disorders were also more prevalent among patients of non-formal employment, the unemployed as well as those presenting in the month of November. Other associations are highlighted in Table 3.

In the final multi-variate model, gender [Prevalence ratio (PR) 1.58 (confidence interval 1.46–1.72)], and Pentecostal faith [PR1.25 (CI1.10–1.42)] remained significant after controlling for the region of the country the patient was from. Other associations are highlighted in Table 4.

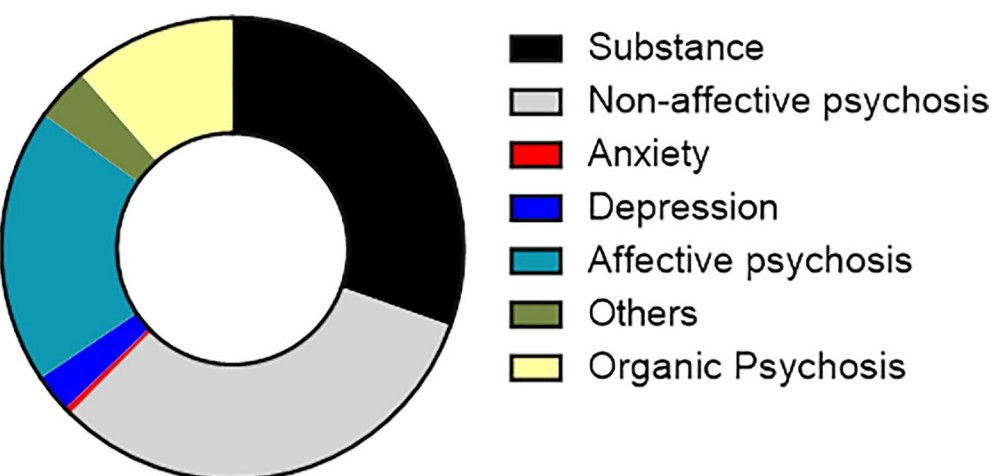

**Fig 2. A pie chart showing the different diagnostic categories for the whole sample.**

## Discussion

### Large burden of psychotic disorders at the National Referral Hospital in Uganda

Over two-thirds (62.7%) of all admissions presenting to the hospital for the first time in 2018 had a psychotic disorder. To our knowledge this is the first published study highlighting the

**Table 2. Background characteristics of the sample of participants classified as having psychosis.**

| Variable | All first time patients (N) | FEP [n(%)] | 95% CI |
|---|---|---|---|
| **Age**[*] | | | |
| ≤ 29 | 757 | 459 (60.6) | 57.1–64.1 |
| > 29 | 674 | 436 (64.7) | 61.0–68.2 |
| **Gender** | | | |
| Male | 930 | 486 (52.3) | 49.0–55.5 |
| Female | 549 | 442 (80.51) | 77.0–83.6 |
| **Religion** | | | |
| Protestant | 404 | 230 (56.9) | 52.0–61.7 |
| Catholic | 407 | 261 (64.1) | 59.3–68.7 |
| Moslem | 242 | 143 (59.1) | 52.8–65.1 |
| Seventh day Adventist | 29 | 17 (58.6) | 40.0–75.0 |
| Pentecostal/Born again | 123 | 95 (77.2) | 69.0–83.8 |
| Other religions | 46 | 29 (63.0) | 48.2–75.8 |
| **Occupation** | | | |
| Student | 89 | 43 (48.3) | 38.1–58.7 |
| Formal | 108 | 63 (58.3) | 48.8–67.3 |
| Non-formal | 270 | 174 (64.4) | 58.5–69.9 |
| Unemployed | 922 | 582 (63.1) | 60.0–66.2 |
| **Region** | | | |
| Central | 1,093 | 675 (61.8) | 58.8–64.6 |
| Eastern | 102 | 67 (65.7) | 55.9–74.3 |
| Northern | 30 | 17 (56.7) | 38.5–73.2 |
| Western | 143 | 93 (65.0) | 56.8–72.4 |

[*]Median used for age categories as age was skewed.

**Table 3. Bivariate analysis of the association between patients with a psychosis diagnosis and different sociodemographic variables.**

| Variable | Total (N) | FEP Prevalence n(%) | Prevalence ratio | 95% CI | P-value |
|---|---|---|---|---|---|
| **Age** | | | | | |
| ≤ 29 | 757 | 459 (60.6) | 1 | 0.98–1.16 | 0.113 |
| > 29 | 674 | 436 (64.7) | 1.07 | | |
| **Gender** | | | | | |
| Male | 930 | 486 (52.3) | 1.00 | **1.43–1.66** | **< 0.001** |
| Female | 549 | 442 (80.51) | 1.54 | | |
| **Religion** | | | | | |
| Protestant | 404 | 230 (56.9) | 1.00 | | |
| Catholic | 407 | 261 (64.1) | 1.13 | **1.01–1.26** | **0.037** |
| Moslem | 242 | 143 (59.1) | 1.04 | 0.91–1.19 | 0.588 |
| Seventh day Adventist | 29 | 17 (58.6) | 1.03 | 0.75–1.41 | 0.857 |
| Pentecostal/Born again | 123 | 95 (77.2) | 1.36 | **1.19–1.54** | **< 0.001** |
| Other religions | 46 | 29 (63.0) | 1.11 | 0.87–1.40 | 0.399 |
| **Occupation** | | | | | |
| Student | 89 | 43 (48.3) | 1.00 | | |
| Formal | 108 | 63 (58.3) | 1.21 | 0.92–1.58 | 0.168 |
| Non-formal | 270 | 174 (64.4) | 1.33 | **1.06–1.68** | **0.015** |
| Unemployed | 922 | 582 (63.1) | 1.31 | **1.05–1.63** | **0.018** |
| **Region** | | | | | |
| Central | 1,093 | 675 (61.8) | 1.00 | | |
| Eastern | 102 | 67 (65.7) | 1.06 | 0.92–1.23 | 0.414 |
| Northern | 30 | 17 (56.7) | 0.92 | 0.67–1.26 | 0.594 |
| Western | 143 | 93 (65.0) | 1.05 | 0.93–1.20 | 0.4321 |

large burden of psychotic disorders in the Ugandan setting among patients presenting for the first time at a mental facility. The retrospective study design limits our interpretation of why there is greater prevalence of psychotic disorders over non-psychotic disorders at initial presentation to the hospital. We submit that future studies may focus on culture and ethnicity to get a better understanding of this large burden of psychotic disorders in this setting. Culture and ethnicity play an important role in symptom presentation, care seeking and access to

**Table 4. Multivariate analysis of the association between FEP and selected exposures.**

| Variable | Prevalence ratio | 95% CI | P-value |
|---|---|---|---|
| **Age** | | | |
| ≤ 29 | 1.00 | 0.92–1.09 | 0.971 |
| > 29 | 0.99 | | |
| **Gender** | | | |
| Male | 1.00 | 1.46–1.72 | **< 0.001** |
| Female | 1.58 | | |
| **Religion** | | | |
| Protestant | 1.00 | | |
| Catholic | 1.11 | 1.00–1.24 | 0.050 |
| Moslem | 1.04 | 0.91–1.18 | 0.603 |
| Seventh day Adventist | 1.00 | 0.74–1.36 | 0.857 |
| Pentecostal/Born again | 1.25 | 1.10–1.42 | **0.001** |
| Other religions | 1.14 | 0.87–1.48 | 0.340 |

health services [40, 41]. Previous literature has also highlighted the preference for alternative and complementary therapies for the initial management of psychotic disorders in this setting [24, 25, 27, 28]. Previous literature by Abbo et al (2009) highlighted that patients are more likely to use both African traditional therapies and biomedicine if the patient has a severe illness or poor global functioning [24]. It is therefore possible that the patients coming to the hospital are the ones who were very ill and generally disruptive in the communities in which they lived. Ethnicity has a strong association to genetic risk which is a key biological risk factor for psychotic disorders [42, 43]. Although psychotic disorders were not found to be more prevalent in any particular ethnic grouping or region of origin, it is important to note that Uganda is one of the most ethnically diverse societies in the world [44]. The participants in this study represented more than 30 different tribes (see S1 Dataset). It would therefore require larger sample sizes to determine an association between a specific ethnicity and onset of psychotic disorders. Currently a large genetic study is underway in Uganda to try and determine the genetic risk for psychotic disorders [45].

## Mental health service requirements for patients with psychotic disorders

The burden for psychotic disorders was greater than that for mood disorders as well as substance use disorders. This suggests that there may be benefit in introducing specialised early intervention services for psychotic disorders at the hospital. Specialised services for psychotic disorders especially at the first episode of psychosis usually lead to better outcomes for patients [46–49]. Currently the hospital has specialised services for substance use disorders, and it would be important to determine the benefit of similar services for psychotic disorders. Future work on necessary components for an early intervention psychosis clinic as well as cost benefit analyses of such a program are recommended [13, 22, 49, 50]. It is also known and often observed that psychotic disorders tend to present with aggression and violence injuring staff and fellow patients [51, 52]. Acute psychiatric units or psychiatric intensive care units have been shown to be especially effective in containing such potentially dangerous behaviour [51], hence calling for such care facilities as useful additions to mental hospitals as opposed to just locked seclusion rooms as is the practice at this facility [51, 52].

## Age of initial presentation with a first episode of psychosis

The low numbers of patients presenting to the hospital younger than 18 years of age requires further review to ensure that it is a case of late onset of illness and not a long duration of untreated illness. The course of psychotic disorders is characterised by a psychosis prodrome before onset of illness usually in the late teens or early adulthood [47, 53]. That most of our patients present outside this age range may imply that either the onset of psychosis is late in this population or that there is a long duration of untreated psychosis (DUP). Both findings have public health significance. Long durations of untreated psychosis have been reported in Sub-Saharan Africa compared to high income countries and have been associated with poorer outcomes [54–56]. Late onset of psychotic illnesses is often associated with a less severe course and better outcomes [57]. This is important for future intervention programs given that both DUP and late onset illness psychosis have different outcome trajectories [14, 53, 58].

## Gender and initial presentation to care with psychotic disorders

Females were more likely to present to the hospital than males with a psychotic illness. The incidence of psychotic disorders is higher in males than females in previous literature [18–22]. Greater prevalence among the female gender might be due to the difference in care seeking between males and females rather than greater incidence in the community. This, however,

would need confirmation with longitudinal studies. It is also important to note that it is unlikely that a patient with psychosis brought themselves to the hospital. Further studies are therefore required to understand why there is preference for bringing females to the hospital than males.

### Religion and initial presentation to care with a psychotic disorder

Psychotic disorders were more prevalent among people of the Pentecostal faith. It is important to clarify that this finding does not mean that people of this faith are more at risk for psychotic disorders. Rather the findings suggest that people of Pentecostal faith with psychotic disorders were more likely than other faiths to seek care from the national referral and psychiatric hospital. Another plausible explanation might be due to explanatory models for mental illness in our setting characterised by beliefs in supernatural causations of psychotic disorders [59]. This may make patients resort to this faith because of its supposed ability to heal mental disorders through prayer hence leading to more psychotic cases there eventually presenting to the hospital [60, 61]. In an ongoing mixed methods study we hope to clarify on this observation by describing the duration of participants in an assigned religious group or changes in religious affiliation after onset of a first episode of psychotic illness [62].

### Limitations of the study

A major limitation of the study was its retrospective study design which could cause information bias. The information however collected was primarily on sociodemographic characteristics which are not usually prone to bias. Also, failure to confirm the diagnoses with a standardized tool could lead to misclassification bias. However, Butabika is a national referral hospital with expertise in mental health care service provision and the diagnoses were made by qualified psychiatrists; so, we were fairly confident in the diagnoses made.

### Conclusion

There seems to be a large burden of psychotic disorders (67%) among patients presenting to the national psychiatric hospital in Uganda for the first time. Many of the participants were female calling for further studies to understand this phenomenon in our setting. More studies are also needed to define the duration of untreated psychosis in this population given that most of the first time patients were older than the normal onset for psychotic disorders. Finally, there may be benefits in introducing specialised intervention services for psychotic disorders at the national referral hospital in the form of specialised early intervention services as well as "safe wards models" as acute psychiatric units or psychiatric intensive care units at such large mental health facilities.

### Supporting information

**S1 Dataset.**
(ZIP)

### Acknowledgments

We acknowledge the patients who presented to the hospital for the first time. Dr. Linnet Ongeri of Kenya Medical Research Institute gave invaluable guidance on the manuscript for which we are grateful.

## Author Contributions

**Conceptualization:** Emmanuel Kiiza Mwesiga, Noeline Nakasujja, Seggane Musisi.

**Data curation:** Emmanuel Kiiza Mwesiga, Annet Nanyonga, Joy Louise Gumikiriza, Paul Bangirana.

**Formal analysis:** Emmanuel Kiiza Mwesiga, Paul Bangirana, Dickens Akena.

**Funding acquisition:** Emmanuel Kiiza Mwesiga, Noeline Nakasujja, Seggane Musisi.

**Investigation:** Emmanuel Kiiza Mwesiga.

**Methodology:** Emmanuel Kiiza Mwesiga, Noeline Nakasujja.

**Project administration:** Emmanuel Kiiza Mwesiga, Juliet Nakku.

**Resources:** Noeline Nakasujja, Juliet Nakku, Annet Nanyonga.

**Supervision:** Emmanuel Kiiza Mwesiga, Juliet Nakku, Joy Louise Gumikiriza.

**Validation:** Emmanuel Kiiza Mwesiga, Juliet Nakku.

**Writing – original draft:** Emmanuel Kiiza Mwesiga, Joy Louise Gumikiriza.

**Writing – review & editing:** Emmanuel Kiiza Mwesiga, Noeline Nakasujja, Juliet Nakku, Annet Nanyonga, Joy Louise Gumikiriza, Paul Bangirana, Dickens Akena, Seggane Musisi.

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
