## [Decision Letter · Decision Letter 0]

2 Dec 2019

PONE-D-19-16103

One year prevalence of psychotic disorders among first treatment contact patients at the National Psychiatric referral and teaching hospital in Uganda

PLOS ONE

Dear Dr. Mwesiga,

Thank you for submitting your manuscript to PLOS ONE. After careful consideration, we feel that it has merit but does not fully meet PLOS ONE’s publication criteria as it currently stands. Therefore, we invite you to submit a revised version of the manuscript that addresses the points raised during the review process.

We would appreciate receiving your revised manuscript by Jan 16 2020 11:59PM. To enhance the reproducibility of your results, we recommend that if applicable you deposit your laboratory protocols in protocols.io, where a protocol can be assigned its own identifier (DOI) such that it can be cited independently in the future. For instructions see: http://journals.plos.org/plosone/s/submission-guidelines#loc-laboratory-protocols

We look forward to receiving your revised manuscript.

Kind regards,

Sphiwe Madiba, DrPH

Academic Editor

PLOS ONE

Journal Requirements:

Reviewers' comments:

Reviewer's Responses to Questions

**Comments to the Author**

1. Is the manuscript technically sound, and do the data support the conclusions?

Reviewer #1: Yes

Reviewer #2: No

2. Has the statistical analysis been performed appropriately and rigorously? 

Reviewer #1: Yes

Reviewer #2: Yes

3. Have the authors made all data underlying the findings in their manuscript fully available?

Reviewer #1: Yes

Reviewer #2: No

4. Is the manuscript presented in an intelligible fashion and written in standard English?

Reviewer #1: Yes

Reviewer #2: Yes

5. Review Comments to the Author

Reviewer #1: Thank you for the opportunity to review this paper.

It is a necessary piece of research and has been written fairly well. However, there are some concerns that need to be addressed before proceeding to the next level.

1. The authors have mentioned the bed capacity of the study site but have not described the bed occupancy. This would be helpful in putting the information provided into context

2. It is not clear how diagnosis in the patient charts was confirmed since there are many cadres (with varying expertise) involved in assigning a diagnosis.

3. It is not clear if the files reviewed were for outpatients or inpatients or both.

4. How many patients does the hospital see annually in outpatient and inpatient units/wards?

5. Most patients become religious as a way of coping with the unusual psychiatric experiences as opposed to change of religion. Was the religious affiliation confirmed by duration in the assigned religious group?

6. The study site/hospital is a referral hospital but rhe study refers to first contact. How many of the patients were accessing the hospital for the first time and how many of them had been referred from the lower health units?

7. In the results section, the prevalence is recorded as 62.7% but in discussion, it is 67%. Which is the correct prevalence for psychotic disorders?

8. Low numbers of patients below 18years should be linked with duration of psychotic symptoms before speculating about late presentation to hospital. Otherwise, it would be a speculation.

9. Which diagnostic criteria were used to diagnose psychotic disorders? This is not clear from the write up.

Reviewer #2: Thank you very much for the opportunity to review the manuscript with the title “One year prevalence of psychotic disorders among first treatment contact patients at the National Psychiatric referral and teaching hospital in Uganda”. Although the findings of the study are interesting, the following points should be addressed to help improved its quality to warrant any publication.

1. The introduction and the rationale for conducting the study was explicitly stated by the authors. In line 56, second paragraph in the introduction, the authors should provide the prevalence rates for anxiety, mood and substance used disorders and the context where this figures apply. This is important as the author made specific reference to these rates in the discussion section.

2. The authors should indicate the specific socio-demographic information that was collected and more importantly the psychological measures that were used for diagnosing the various psychotic disorders.

3. The authors used 0.2 as the level of significance for possible inclusion of variables into the multivariate analysis. It is unclear why the authors set such a high margin contrary to the default level of 0.05 which was used later on in the subsequent analysis. A strong justification for setting 20% margin of error should be provided.

4. In the methods section, it also important for the authors to inform the reader how the various variables in the study were scored? Was there any re-coding of variables?, This information should be provided.

5. The authors presented age with only those categories: ≤ 29 years and > 29 years. Was there any justification of this re-categorisation when the authors originally presented more than 2 ranges of ages as presented under the burden of psychotic disorders. For example, it would be informative to known how many of the participants were in their original categorized age groups before re-categorization

6. The full meaning of FEP should be provided at least in the first instance before using the acronym in subsequent citations

7. Some of the key findings of the study were not discussed. This manuscript did not assessed the role of culture in the presentation to care with psychotic disorders, yet a huge paragraph was dedicated to this. It is important for the authors to strictly limit themselves to discussing only the key findings of this study.

6. PLOS authors have the option to publish the peer review history of their article (what does this mean?). If published, this will include your full peer review and any attached files.

Reviewer #1: No

Reviewer #2: No

---

## [Author Response · Author response to Decision Letter 0]

17 Dec 2019

Journal Requirements:

Response: The files have been renamed as 

i) “Response to reviewers” file.

ii) "Revised Article with Changes Highlighted” file.

iii) “Manuscript” file.

Response: We have uploaded a minimal anonymized data set necessary to replicate our study findings as Supporting Information files. This has been reflected in the data availability statement on page 9 lines 289 to 287. We have also added it to our cover letter.

Response: The title in the manuscript has been changed to reflect the one on the online submission. It now reads, “One year prevalence of psychotic disorders among first treatment contact patients at the National Psychiatric Referral and Teaching Hospital in Uganda.”

Reviewer #1: 

1. The authors have mentioned the bed capacity of the study site but have not described the bed occupancy. This would be helpful in putting the information provided into context

Response: At the beginning of the results section on page 4 line 150 the bed occupancy has been provided as 149%. The complete full statement reads as follows; “Between January 1st, 2018 and December 31st, 2018; 43,870 patients were seen in the outpatient mental health clinic and 10,578 patients were admitted with a bed occupancy rate of 149%. 1685 patients accessed services from Butabika for the first time and made up the study sample.”

2. It is not clear how diagnosis in the patient charts was confirmed since there are many cadres (with varying expertise) involved in assigning a diagnosis.

Response: The failure to confirm the diagnosis was a limitation of the study and was reflected in the limitations section. We would also like to add that we are soon to submit a manuscript that looked at the concordance between clinician diagnoses at admission and the standardised MINI international neuropsychiatric inventory for DSM 5 disorders. In that publication we will report a concordance of 50% between clinician and MINI diagnoses.

3. It is not clear if the files reviewed were for outpatients or inpatients or both.

Response: We have added a line on page 3 lines 122 to 124 in order to clarify on this. The statement reads as follows, “The files reviewed were for participants who presented for the first time at the hospital irrespective of whether they were outpatients and returned home or admitted as in-patients for further management.”

4. How many patients does the hospital see annually in outpatient and inpatient units/wards?

Response: This has been answered in item 1 above. The numbers for 2018 are found on page 4 lines 148 to 151.

5. Most patients become religious as a way of coping with the unusual psychiatric experiences as opposed to change of religion. Was the religious affiliation confirmed by duration in the assigned religious group?

Response: No, the duration in the assigned religious group was not confirmed in this study as this was a retrospective chart review. There is however an ongoing study that hopes to describe this observation. We have reported this on page 9 lines 256 to 258 and reads as follows, “In an ongoing mixed methods study we hope to clarify on this observation by describing the duration of participants in an assigned religious group or changes in religious affiliation after onset of a first episode of psychotic illness (1).”

6. The study site/hospital is a referral hospital but the study refers to first contact. How many of the patients were accessing the hospital for the first time and how many of them had been referred from the lower health units?

Response: All the patients included in the final analysis were accessing the hospital for the first time in the calendar year. Source of referral is unfortunately not collected in the initial face sheet records of the hospital.

7. In the results section, the prevalence is recorded as 62.7% but in discussion, it is 67%. Which is the correct prevalence for psychotic disorders?

Response: This has been corrected to 62.7% in the discussion section on page 7 line 191.

8. Low numbers of patients below 18years should be linked with duration of psychotic symptoms before speculating about late presentation to hospital. Otherwise, it would be a speculation.

Response: Thank you for this correction. We have re written the section on page 8 lines 227 to 238 and now reads as follows:

Age of initial presentation with a first episode of psychosis: The low numbers of patients presenting to the hospital younger than 18 years of age requires further review to ensure that it is a case of late onset of illness and not a long duration of untreated illness. The course of psychotic disorders is characterised by a psychosis prodrome before onset of illness usually in the late teens or early adulthood (46, 52). That most of our patients present outside this age range may imply that either the onset of psychosis is late in this population or that there is a long duration of untreated psychosis (DUP). Both findings have public health significance. Long durations of untreated psychosis have been reported in Sub-Saharan Africa compared to high income countries and have been associated with poorer outcomes (53-55). Late onset of psychotic illnesses is often associated with a less severe course and better outcomes (56). This is important for future intervention programs given that both DUP and late onset illness psychosis have different outcome trajectories (14, 52, 57). 

9. Which diagnostic criteria were used to diagnose psychotic disorders? This is not clear from the write up.

Response: The clinicians’ diagnoses were made by the hospital clinicians who often a trained to current guidelines. Currently the hospital uses the DSM 5 criteria. This has been reported on page 3 lines 115 to 116 and reads as follows, “Diagnoses as well as criteria for severity are made based on the Diagnostic and Statistical Manual of Mental disorders 5th edition (DSM-5) (2).

Reviewer #2: 

1. The introduction and the rationale for conducting the study was explicitly stated by the authors. In line 56, second paragraph in the introduction, the authors should provide the prevalence rates for anxiety, mood and substance used disorders and the context where this figures apply. This is important as the author made specific reference to these rates in the discussion section.

Response: A line reflecting the limited data in low resource settings was added on page 2 lines 57 to 60 and reads, “In a previous review by Steel et al 2014, the period prevalence of common mental disorders like anxiety, mood and substance use disorders was found to be lower in low resource countries than high income countries (3).”

2. The authors should indicate the specific socio-demographic information that was collected and more importantly the psychological measures that were used for diagnosing the various psychotic disorders.

Response: We added some information to the methods section on page 3 lines 110 to 116 that now reads as follows:

The patient sociodemographic variables including age, gender, ethnicity, religion, occupation and home district are recorded in the file before the patient is sent to see a clinician. These were the sociodemographic characteristics abstracted for in the chart review. The clinician then makes a diagnosis, and a decision of whether to treat the patient as an out-patient or send them to admission in one of the units described above. Diagnoses as well as criteria for severity are made based on the Diagnostic and Statistical Manual of Mental disorders 5th edition (DSM-5) (2).

3. The authors used 0.2 as the level of significance for possible inclusion of variables into the multivariate analysis. It is unclear why the authors set such a high margin contrary to the default level of 0.05 which was used later on in the subsequent analysis. A strong justification for setting 20% margin of error should be provided.

Response: This was an error as the level for bivariate analysis was 0.2 and that for multivariate analysis was 0.05. This has been corrected and the full description on page 4 lines 142 to 146 and now reads, 

Variables with a level of significance less than 0.2 were included in the bivariate analysis. However, region of origin was assessed for any possible confounding effects as ethnicity has been shown to have a genetic biological risk factor for psychotic disorders. At multi-variate analysis a level of significance of less than 0.05 was used to test for significance between different exposures and FEP.

4. In the methods section, it also important for the authors to inform the reader how the various variables in the study were scored? Was there any re-coding of variables?, This information should be provided.

Response: This information has been provided for the age variable. On page 4 lines 138 to 139 the following line was added, “As the age of the participants was skewed, this was recoded to those with ages less than or equal to the median and those greater than the median.”

5. The authors presented age with only those categories: ≤ 29 years and > 29 years. Was there any justification of this re-categorisation when the authors originally presented more than 2 ranges of ages as presented under the burden of psychotic disorders. For example, it would be informative to known how many of the participants were in their original categorized age groups before re-categorization

Response: The median was used to categorise the age as the data was skewed. We have however shown the different categories in table 1 as follows:

Variable Number (N) Percentage (%)

Age 

18-24 459 28.8

25-29 377 23.6

30-36 371 23.3

37-47 231 14.5

48-53 81 5.1

6. The full meaning of FEP should be provided at least in the first instance before using the acronym in subsequent citations

Response: This has been defined in the first introduction paragraph on page 2 lines 47 to 49. It now reads as follows; “Correct management at initial presentation of psychotic disorders -operationally defined as the first episode of psychosis (FEP); has been associated with lower relapse rates, greater functional recovery and improved quality of life (4, 5).” 

In the methods section on page 3 lines 126 to 127 it has been reiterated as follows; “As these participants with psychotic disorders were presenting to the clinic for the first time they were classified as having a first episode of psychosis (FEP).”

7. Some of the key findings of the study were not discussed. This manuscript did not assess the role of culture in the presentation to care with psychotic disorders, yet a huge paragraph was dedicated to this. It is important for the authors to strictly limit themselves to discussing only the key findings of this study.

Response: 

a) We acknowledge that culture and ethnicity per se were not presented in the results, but this data had been collected but not presented. We added a line to show the ethnic diversity of the study population at the beginning of the results section on page 4 lines 154 to 156 that reads as follows, “There was wide representation of different ethnicities in the sample with over 25 different tribes presenting to the hospital.” 

b) The initial discussion on culture and ethnicity has been moved to an earlier part of the discussion to postulate why more participants present with psychotic disorders compared to other mental illnesses in this setting. Page 7 Lines 190 to 205 and page 8 lines 206 to 212 now read as follows;

Large burden of psychotic disorders at the National referral hospital in Uganda: Over two-thirds (67%) of all admissions presenting to the hospital for the first time in 2018 had a psychotic disorder. To our knowledge this is the first published study highlighting the large burden of psychotic disorders in the Ugandan setting among patients presenting for the first time at a mental facility. The retrospective study design limits our interpretation of why there is greater prevalence of psychotic disorders over non-psychotic disorders at initial presentation to the hospital. We submit that future studies may focus on culture and ethnicity to get a better understanding of this large burden of psychotic disorders in this setting. Culture and ethnicity play an important role in symptom presentation, care seeking and access to health services (6, 7). Previous literature has also highlighted the preference for alternative and complementary therapies for the initial management of psychotic disorders in this setting (23, 24, 26, 27). Previous literature by Abbo et al (2009) highlighted that patients are more likely to use both African traditional therapies and biomedicine if the patient has a severe illness or poor global functioning (8). It is therefore possible that the patients coming to the hospital are the ones who were very ill and generally disruptive in the communities in which they lived. Ethnicity has a strong association to genetic risk which is a key biological risk factor for psychotic disorders (9, 10). Although psychotic disorders were not found to be more prevalent in any particular ethnic grouping or region of origin, it is important to note that Uganda is one of the most ethnically diverse societies in the world (11). The participants in this study represented more than 30 different tribes. It would therefore require larger sample sizes to determine an association between a specific ethnicity and onset of psychotic disorders. Currently a large genetic study is underway in Uganda to try and determine the genetic risk for psychotic disorders (12).”

c) Catholic religion had a level of significance of 0.05 and therefore one could not reject the null hypothesis. This therefore was not discussed. The p value is not marked as bold in the revised manuscript.

References.

1. Mwesiga EK, Nakasujja N, Ongeri L, Semeere A, Loewy R, Meffert S. A cross-sectional mixed methods protocol to describe correlates and explanations for a long duration of untreated psychosis among patients with first episode psychosis in Uganda. BMJ open. 2019;9(7):e028029.

2. Association AP. DSM 5: American Psychiatric Association; 2013.

3. Steel Z, Marnane C, Iranpour C, Chey T, Jackson JW, Patel V, et al. The global prevalence of common mental disorders: a systematic review and meta-analysis 1980-2013. Int J Epidemiol. 2014;43(2):476-93.

4. Marshall M, Lockwood A, Lewis S, Fiander M. Essential elements of an early intervention service for psychosis: the opinions of expert clinicians. BMC Psychiatry. 2004;4:17.

5. Marshall M, Rathbone J. Early intervention for psychosis. (1469-493X (Electronic)).

6. Maraj A, Anderson KK, Flora N, Ferrari M, Archie S, McKenzie KJ. Symptom profiles and explanatory models of first-episode psychosis in African-, Caribbean- and European-origin groups in Ontario. Early Interv Psychiatry. 2017;11(2):165-70.

7. Singh SP, Brown L, Winsper C, Gajwani R, Islam Z, Jasani R, et al. Ethnicity and pathways to care during first episode psychosis: the role of cultural illness attributions. BMC Psychiatry. 2015;15:287.

8. Abbo C, Ekblad S, Waako P, Okello E, Musisi S. The prevalence and severity of mental illnesses handled by traditional healers in two districts in Uganda. Afr Health Sci. 2009;1(9):S16-22.

9. Busby GB, Band G, Si Le Q, Jallow M, Bougama E, Mangano VD, et al. Admixture into and within sub-Saharan Africa. eLife. 2016;5.

10. Stevenson A, Akena D, Stroud RE, Atwoli L, Campbell MM, Chibnik LB, et al. Neuropsychiatric Genetics of African Populations-Psychosis (NeuroGAP-Psychosis): a case-control study protocol and GWAS in Ethiopia, Kenya, South Africa and Uganda. BMJ open. 2019;9(2):e025469.

11. Alesina AFaE, William and Devleeschauwer, Arnaud and Kurlat, Sergio and Wacziarg, Romain T.,. Fractionalization (June 2002). Harvard Institute Research Working Paper No. 1959.

12. Anne Stevenson, Dickens Akena, Rocky E Stroud, Lukoye Atwoli, Megan M Campbell, Lori B Chibnik, et al. Neuropsychiatric Genetics of African Populations-Psychosis (NeuroGAPPsychosis): a case-control study protocol and GWAS in Ethiopia, Kenya, South Africa and Uganda. BMJ open. 2019:e025469. doi:10.1136/bmjopen-2018-.

---

## [Decision Letter · Decision Letter 1]

8 Jan 2020

One year prevalence of psychotic disorders among first treatment contact patients at the National Psychiatric referral and teaching hospital in Uganda

PONE-D-19-16103R1

Dear Dr. Mwesiga,

We are pleased to inform you that your manuscript has been judged scientifically suitable for publication and will be formally accepted for publication once it complies with all outstanding technical requirements.

With kind regards,

Sphiwe Madiba, DrPH

Academic Editor

PLOS ONE

Reviewers' comments:

Reviewer's Responses to Questions

**Comments to the Author**

1. If the authors have adequately addressed your comments raised in a previous round of review and you feel that this manuscript is now acceptable for publication, you may indicate that here to bypass the “Comments to the Author” section, enter your conflict of interest statement in the “Confidential to Editor” section, and submit your "Accept" recommendation.

Reviewer #1: All comments have been addressed

Reviewer #2: All comments have been addressed

2. Is the manuscript technically sound, and do the data support the conclusions?

Reviewer #1: Yes

Reviewer #2: Yes

3. Has the statistical analysis been performed appropriately and rigorously? 

Reviewer #1: Yes

Reviewer #2: Yes

4. Have the authors made all data underlying the findings in their manuscript fully available?

Reviewer #1: Yes

Reviewer #2: No

5. Is the manuscript presented in an intelligible fashion and written in standard English?

Reviewer #1: Yes

Reviewer #2: Yes

6. Review Comments to the Author

Reviewer #1: I am satisfied that my previous review comments have been adequately addressed. Thank you for the good job done.

Reviewer #2: The authors have addressed all the comments I raised in the previous draft. The content of the manuscript has improved considerably

7. PLOS authors have the option to publish the peer review history of their article (what does this mean?). If published, this will include your full peer review and any attached files.

Reviewer #1: Yes: Dr. Godfrey Zari Rukundo (MBChB, MMed, PhD), Senior Lecturer, Department of Psychiatry, Mbarara University of Science and Technology, Uganda

Reviewer #2: No

---

## [Editor Report · Acceptance letter]

10 Jan 2020

PONE-D-19-16103R1 

One year prevalence of psychotic disorders among first treatment contact patients at the National Psychiatric referral and teaching hospital in Uganda 

Dear Dr. Mwesiga:

I am pleased to inform you that your manuscript has been deemed suitable for publication in PLOS ONE. Congratulations! Your manuscript is now with our production department. 

With kind regards,

on behalf of

Dr. Sphiwe Madiba 

Academic Editor

PLOS ONE